# Role of the Microenvironment in Regulating Normal and Cancer Stem Cell Activity: Implications for Breast Cancer Progression and Therapy Response

**DOI:** 10.3390/cancers11091240

**Published:** 2019-08-24

**Authors:** Vasudeva Bhat, Alison L. Allan, Afshin Raouf

**Affiliations:** 1Department of Immunology, Max Rady Faculty of Health Sciences, University of Manitoba, Winnipeg, MB R3E 0T5, Canada; 2Research Institute in Oncology and Hematology, CancerCare Manitoba, Winnipeg, MB R3E 0V9, Canada; 3London Regional Cancer Program, London Health Science Centre, London, ON N6A 5W9, Canada; 4Department of Anatomy & Cell Biology and Oncology, Western University, London, ON N6A 3K7, Canada

**Keywords:** microenvironment, mammary stem cells, breast cancer stem cells, hypoxia, immune cells, cytokines

## Abstract

The epithelial cells in an adult woman’s breast tissue are continuously replaced throughout their reproductive life during pregnancy and estrus cycles. Such extensive epithelial cell turnover is governed by the primitive mammary stem cells (MaSCs) that proliferate and differentiate into bipotential and lineage-restricted progenitors that ultimately generate the mature breast epithelial cells. These cellular processes are orchestrated by tightly-regulated paracrine signals and crosstalk between breast epithelial cells and their tissue microenvironment. However, current evidence suggests that alterations to the communication between MaSCs, epithelial progenitors and their microenvironment plays an important role in breast carcinogenesis. In this article, we review the current knowledge regarding the role of the breast tissue microenvironment in regulating the special functions of normal and cancer stem cells. Understanding the crosstalk between MaSCs and their microenvironment will provide new insights into how an altered breast tissue microenvironment could contribute to breast cancer development, progression and therapy response and the implications of this for the development of novel therapeutic strategies to target cancer stem cells.

## 1. Introduction

Adult tissue regeneration and maintenance are mainly regulated by continual turnover of mature cells. This process is mediated by the presence of tissue-specific stem cells, the functions of which depend on both intrinsic and extrinsic molecular signals. Extrinsic signals from the extracellular environment can activate intracellular signaling required for the expression of genes related to self-renewal, proliferation, differentiation and cell-fate commitment of stem cells. Adult stem cells reside in a specific tissue microenvironment composed of cellular components such as stromal fibroblasts, tissue-specific mature cells, immune cells, adipose and endothelial cells. The non-cellular portion of the stem cell microenvironment includes extracellular matrix components, growth factors and cytokines. Interactions with their surrounding tissue microenvironment provides stem cells with favorable conditions to either self-renew, proliferate, or differentiate into progenitor cells [1,2]. Such a notion is supported by observations indicating that primary cells, once isolated from their native microenvironment, exhibit altered proliferation and differentiation potentials that can be reinstated by controlling their microenvironment in ex vivo cultures [3,4].

Studies performed using mouse models have also provided data reinforcing the observations made using primary human cells. For example, while the injection of carcinoma cells into blastocysts resulted in the generation of genetically normal mice, the subcutaneous injection of the same cells resulted in the development of teratomas [5]. Subsequent studies using Rous sarcoma virus that contained the oncogene pp60src demonstrated that injection of the virus in the wing of a chick resulted in a tumor, while injection of the virus into the chick embryo failed to form tumors [6,7,8]. These studies suggest that the tissue microenvironment can play either a tumor-suppressive or a tumor-promoting role depending on the physiological context. 

The stem cell microenvironment, also known as the stem cell ‘niche’, has been extensively studied with respect to its critical role in regulating hematopoietic stem cell self-renewal and differentiation leading to the maintenance of the human hematopoietic system. In addition, the role of the non-cellular components of the stem cell microenvironment has also been extensively studied using animal models [9]. For example, the role of extracellular matrix proteins such as β1 integrins in skin [10,11]; osteopontin in the hematopoietic system [12,13] and tenascin C in the nervous system [14,15] have all been shown to play an essential role in regulating tissue-specific stem cell functions.

## 2. Normal Mammary Tissue Microenvironment and Function

### 2.1. Cellular Composition of the Normal Mammary Gland

The mammary gland is an intricate network of interconnected ducts and alveolar structures. These structures are composed of both luminal and myoepithelial cells. In ducts, luminal cells are organized to form hollow tubes that are surrounded by a continuous layer of myoepithelial cells, whereas in alveoli, luminal cells are arranged to form clusters of grape-like structures that are surrounded by non-contiguous myoepithelial cells, allowing luminal cells to be in constant contact with the surrounding stroma [16,17]. In contrast to the mouse mammary gland where the alveolar structures are surrounded mostly by adipose cells, in human breast tissue the bilayered ducts and alveolar structures are surrounded by a basement membrane composed of laminin and collagen. During pregnancy, luminal cells within the alveoli can further differentiate into milk-producing cells under the influence of prolactin. The myoepithelial cells are essential for milk ejection into the ducts by contracting in the presence of oxytocin [18,19,20]. These epithelial cells double in number during each estrous cycle [21,22,23]. Interestingly, during pregnancy and lactation, the epithelial content of the breast tissue also expands by up to nine times the original cell numbers. During pregnancy in mice, a 27-fold increase in epithelial cell number has been reported [24]. Post weaning, epithelial cells undergo apoptosis and the gland reverts back to a non-pregnant state through a process known as involution [25,26]. This dynamic process of expansion and regression makes the gland highly regenerative and allows the female breast to support multiple pregnancies. 

This extensive regenerative potential of the mammary gland is due to the presence of the primitive mammary stem cells (MaSCs), which can give rise to both luminal and myoepithelial cells that make up the ductal and alveolar structures. For the sake of simplicity, both mouse mammary and human breast stem cells will be referred to as MaSCs in this article. Experimental evidence has demonstrated the highly regenerative capacity of the mammary gland, whereby even a small fragment of mouse mammary structure transplanted into de-epithelized (cleared) mammary fat pads can regenerate the entire mammary gland [27]. In support of this observation, subsequent studies showed that any part of the mammary epithelial tree can produce successful engraftment [28,29,30], suggesting that cells with regenerative capacity are dispersed throughout the mammary tree. Isolation of mouse MaSCs has been made possible through the identification of cell surface markers enabling the study of their proliferation, differentiation, and self-renewal potentials in vitro and in vivo [31,32]. These studies also demonstrate that MaSCs obtained from mouse mammary glands are able to generate bilayered mammary structures containing both luminal and myoepithelial cells [31,32]. However, current evidence suggests that in postnatal mammary gland, the MaSCs are heterogenous in nature and consist of unipotent stem cells capable of generating luminal or myoepithelial cells [33,34,35]. This finding challenges the bipotent property of MaSCs in the postnatal mouse mammary gland. 

Current evidence suggests that human breast MaSCs reside in the ductal structures of the mammary gland [36], although the presence of human MaSCs in other locations in the gland remains unexplored. Xenotransplantation of CD49f^high^EpCAM^low/−^ human breast epithelial cells into mouse renal capsules resulted in generation of mammary structures, albeit at a low frequency [37]. This low regeneration of mammary structures could be either due to the lack of unique markers that provide further enrichment of MaSCs, or due to the lack of an appropriate/favorable microenvironment to facilitate the regenerative ability of human MaSCs in the renal capsule. Notably, the mouse mammary gland and human breast tissue microenvironments are different in their composition. Human breast tissue consists of collagen-rich inter- and intra-lobular stroma which is absent in the mouse mammary gland. In contrast, the mouse mammary gland is mainly made up of an adipose-rich stroma that surrounds the ducts [38]. Generation of humanized mammary mouse models is possible and subcutaneous implantation of human breast tissue in this model can accurately recapitulate the microenvironment of the human breast [39,40].

### 2.2. Components of the Normal Breast Tissue Microenvironment

MaSCs, like other tissue-specific stem cells, reside in a niche (microenvironment) in the breast that consists of different cell types including epithelial cells, fibroblasts, adipocytes, vascular endothelial cells, and immune cells (Figure 1). The breast tissue niche also includes non-cellular components such as basement membrane (BM) and extracellular matrix (ECM) components, growth factors, and cytokines that are vital for cell function. MaSCs have the ability to self-renew, proliferate and/or differentiate to generate mature luminal and the myoepithelial cells of the breast tissue. To this end, the autocrine and paracrine signals initiated by growth factors and cytokines of the niche, the regulatory signals initiated by the matrix components (laminin and collagen in particular), as well as cell-cell interactions within the niche are essential to the regulation of MaSC function [41,42,43,44,45,46,47,48].

In the ducts, the luminal cells are surrounded by a continuous layer of myoepithelial cells, However, in the alveolar structures, the luminal cells are surrounded by discontinuous layer of myoepithelial cells [49,50] allowing the luminal cells of the alveolar structures to interact with and receive signals from the different microenvironment components. Such interactions facilitate the further differentiation of alveolar luminal cells into milk-producing cells. Myoepithelial cells and fibroblasts are capable of secreting important ECM components including fibronectins, laminins, collagens, and proteoglycans that in turn help provide a defined ECM architecture and the necessary signals for highly regulated functions of mature cells, progenitors, and MaSCs within the mammary gland [51]. During the lactation and involution phases of pregnancy, this complex ECM is disrupted and then re-formed based on the action of different proteases such as matrix metalloproteinases (MMPs), as well as the action of immune cells such as eosinophils and mast cells [52,53,54]. During mouse mammary gland development at puberty, these immune cells are recruited near the terminal end buds (similar to TDLUs found in the human breast tissue) and mediate ductal outgrowth and branching morphogenesis by remodeling surrounding matrix [55].

#### 2.2.1. Immune Cells

Immune cells such as macrophages, eosinophils, neutrophils, and mast cells have been shown to play an important role in normal mammary gland development [55,56,57,58,59,60]. Macrophages in particular have been demonstrated to play a vital role in mammary gland development. Transplantation of MaSCs into macrophage-deficient mouse mammary fat pads showed defective mammary reconstitution ability [61], suggesting that the presence of macrophages throughout mammary gland development process is required for the normal functioning of MaSCs. Additional roles of macrophages in regulating mammary gland development via their direct interaction with MaSCs was reported recently [62]. The expression of the Notch receptor ligand Dll1 on MaSCs was shown to interact with Notch3 receptor (Nr3) expressed on adjacent macrophages, resulting in activation of intracellular Notch signaling. This interaction was shown to be necessary for the maintenance of macrophage numbers in the mammary gland, as well as secretion of Wnt ligands such as Wnt3a, Wnt10 and Wnt16 into the MaSC niche. These macrophage-secreted Wnt ligands then utilize positive feedback mechanisms to regulate MaSC activity [62]. Observations by Zeng and Nusse also demonstrated that in mice, Wnt3A-responsive cells were enriched in MaSCs, and that MaSCs exposed to Wnt3A displayed enhanced regenerative ability in vivo [63].

#### 2.2.2. Extracellular Matrix

Extracellular matrix components such as laminin are also known to interact with integrin receptors expressed by stem and progenitor cells, and these interactions transduce signals required for the normal functioning of undifferentiated cells [64,65,66]. It therefore comes as little surprise that cells expressing α6 and α1 integrins display mammary regenerative abilities in vivo [31,32,36,37]. Alpha-1 integrins have been shown to be involved in the proliferation of alveolar progenitors [67] as well as maintenance and regulation of regenerative ability of MaSCs [68]. ECM components have also been shown to regulate the expression of α1 integrins on both human and mouse mammary epithelial cells [69]. Moreover, protein microarray analysis of the ECM revealed that laminin 1 is required for maintenance of bipotential progenitors in a quiescent state, while P-Cadherin is required for myoepithelial cell differentiation [70]. These observations are particularly interesting in light of additional findings that the presence of α6 integrin-expressing bipotent progenitors in laminin-enriched Matrigel results in their proliferation without differentiation, while placing the same cells on collagen-coated plates results in their differentiation into mature luminal and myoepithelial cells [70]. In contrast, luminal cell differentiation appears to be instead dependent on cell-cell contact [70]. These observations identify the ECM as a strong modulator of MaSC and progenitor cell functions during normal mammary gland development.

#### 2.2.3. Stroma

Studies in mouse models have also demonstrated the influence of stromal cells on mammary gland development. Epithelial-stromal cross-talk is necessary for the proper development and maintenance of the mammary gland [43]. A recent study has demonstrated that Gli2 expressing stromal cells secreted paracrine factors (Igf1, Fgf7, Hgf, Wnt2B, and Bmp7) to promote MaSC self-renewal and ductal outgrowth [71]. It has been observed that when mammary epithelium is recombined with salivary gland mesenchyme, the epithelium differentiates into salivary gland structures [45]. Interestingly, xenotransplantation of mouse embryonic skin epidermal cells into the mouse renal capsule along with embryonic mammary mesenchyme of either the rat- or mouse resulted in the generation of bilayered mammary ductal structures. These structures consisted of epithelial cells capable of responding both to estrogen and lactogenic hormones by differentiating into milk producing cells [72]. Additional studies by Boulander et al. demonstrated that non-mammary epithelial stem cells exposed to the mammary gland microenvironment were capable of generating a functional mammary gland that contained cells capable of reconstituting mammary fat pads in serial transplantations [73,74,75]. In vitro models have shown that genes such as HDAC7 that regulate breast epithelial cell proliferation are also capable of reprogramming the extracellular microenvironment [76]. Furthermore, an in vitro 3D Matrigel culture system demonstrated that the regenerative ability of MaSCs was enhanced in presence of fibroblasts, a major stromal component of the breast tissue [77], suggesting the importance of mammary fibroblast in regulating MaSC activity. Taken together, these studies highlight the importance of the stromal microenvironment in defining cell fate and tissue function.

## 3. Breast Tumor Microenvironment

Just as the normal tissue environment plays a critical role in regulating mammary stem/progenitor cell functions, accumulating evidence suggest that the tumor microenvironment (TME) also plays an essential role in regulating cancer stem cell (CSC) activity [78,79,80,81,82,83,84,85,86,87,88,89] and tumor progression. Current evidence indicates that similar to normal breast tissue, breast tumor growth and progression is regulated through hierarchically organized cancer cell populations which are maintained by CSCs that exhibit self-renewal and proliferation potentials [90,91]. The direct experimental evidence demonstrating the transformation of MaSCs into bCSC remains elusive. Interestingly, both normal and cancer stem cells express common markers such as CD44 and ALDH [92,93]. In addition, conserved signaling pathways such as Notch and Wnt that regulate MaSC function (i.e. self-renewal, proliferation, and cell fate determination) are also active in bCSCs [94,95,96]. Molyneux et al, showed that deletion of *BRCA1* in human breast luminal progenitors resulted in basal-like breast cancers on *P53* mutant background [97]. These findings suggest that both normal MaSCs and/or mammary progenitors may have the potential to transform into bCSCs. 

These CSCs are thought to be responsible for tumor recurrence and therapy resistance [98,99,100]. Previously, it was believed that resistance to chemotherapeutic drugs was acquired through accumulation of genetic alterations that generate a heterogeneous population of tumor cells with diverse phenotypes [101,102]. However, the cancer stem cell hypothesis suggests that since CSCs are responsible for maintaining tumor cells, the lack of therapies for specifically targeting these CSCs is responsible for tumor recurrence [103,104,105,106,107,108,109,110]. This issue can be addressed, at least in part, by advances in next generation sequencing (NGS) platforms that have enabled the examination of genomic and transcriptomic changes of tumors at the single cell level [111,112,113,114,115]. Such powerful technology has revealed that tumors (including breast tumors), can undergo a clonal evolution process which is a driving force behind tumor heterogeneity [116,117]. Moreover, comparing therapy-resistant metastatic tumors to matched primary tumors using single-cell genomics has revealed the existence of therapy-resistant clonal cells in the primary tumors; further supporting the role of CSCs in therapy resistance and tumor progression [118]. 

Breast cancer stem cell (bCSC) functions can be influenced by different cytokines and cell types present in the TME, including mesenchymal stem cells (MSCs), cancer associated fibroblasts (CAFs), and tumor associated leukocytes (TILs) (summarized in Table 1) [119]. Interestingly, in addition to the role of the primary TME in regulating bCSC activity, organ-specific microenvironments play an important role in the metastatic process. Previously, Chu et al demonstrated that soluble factors from the lung microenvironment induced chemotactic migration of CD44^+^ALDH^high^ bCSCs, suggesting an interaction between bCSCs and the microenvironment in regulating tissue-specific metastasis [120]. Furthermore, bone-derived osteopontin has been shown to maintain the bCSC phenotype and promote bone metastasis [121]. These observations strongly suggest that the microenvironment is an important modulator of bCSC function including therapy resistance, recurrence and metastasis. Therefore, understanding the interaction between bCSCs and their microenvironment will help in the identification of new therapeutic targets for improved treatment of breast cancer.

### 3.1. Cytokines

In addition to matrix components, the TME contains several non-cellular components including cytokines, chemokines, and growth factors that are secreted by the various cell types that make up the TME. These cytokines can create a chronic inflammatory environment that favors tumor cell survival and disease progression [122,123,124] while at the same time suppressing immune cell functions.

#### 3.1.1. Interleukins 

The cytokine interleukin-6 (IL-6) has been shown to increase the expression of a CD44, a known marker of bCSCs [140], in MCF10A cells expressing tamoxifen induced Src kinase oncoprotein (MCF10A-Scr) [125]. In addition, CD44^high^ MCF10A-Scr cells generated tumors at a higher frequency as compared to CD44^low^ cells in mouse xenografts. When breast cancer cells derived from invasive ductal carcinoma tissues were treated with transformed MCF10A conditioned media, there was a conversion or dedifferentiation of CD44^low^ non-bCSCs to CD44^high^ bCSCs [125]. These observations suggest that extracellular factors present in the TME can play an important role in promoting stemness in breast cancer cells. Interestingly, IL-6-induced stemness in breast cancer cells has been shown to occur by activating the expression of OCT4 gene via the janus kinase/signal transducer and activator of transcription protein 3 (JAK1/STAT3) pathway [126]. In addition, IL-6 has been shown to upregulate jagged 1 (JAG1) and activate JAG1-NOTCH3 signaling, which ultimately results in higher secretion of IL-6 [127]. Thus, autocrine IL-6 signaling can then increase the proliferation and self-renewal potentials of CSCs that exhibit higher expression of NOTCH3 [127]. A recent study showed that presence of the IL-6 superfamily member, oncostatin-M (OM) in the TME upregulates genes related to the CSC phenotype such as SNAIL and CD44 in TNBC cells lines, leading to enhanced tumor formation in vivo. However, this effect of oncostatin-M was inhibited in the presence of IFN-β [86], suggesting that IFN-β could be an effective therapeutic agent against CSCs in TNBC.

Previous reports indicate that ALDH1^+^ bCSCs showed a higher expression of the IL-8 receptor and the CXCR gene [141]. To this end, IL-8 signaling has been associated with enhanced CSC activity and chemoresistance in triple negative breast cancers (TNBCs) [128]. Interestingly, the IL-8-CXCR1 signaling axis has been shown to be important in regulating bCSC function in HER2-positive breast cancers [129]. Patient-derived breast cancer cells demonstrated enhanced mammosphere formation in the presence of IL-8, while inhibition of IL-8-CXCR1 and HER2 signaling impaired mammosphere-forming activity [129]. These findings suggest that the IL-8-CXCR1 signaling axis could be a useful therapeutic target in treating HER2-positive breast cancer patients. Furthermore, inhibition of CXCL1 by reparixin reduced bCSC activity and prevented metastatic spread of breast cancer cells in mouse xenograft models [142]. 

#### 3.1.2. Transforming Growth Factor β and Tumor Necrosis Factor α

The role of transforming growth factor β (TGFβ) in regulating tumor cell proliferation, metastasis, and remodeling of the TME has also been well documented [143]. Very recently, Katsuno et al. demonstrated that prolonged exposure of human breast epithelial cells to TGFβ enhanced the epithelial-to-mesenchymal transition (EMT) phenotype and increased the number of CD44^high^CD24^low^ cell population [130]. Using a mouse breast cancer model, it was also recently demonstrated that TGFβ mediated homing of human bone-marrow derived stem cells to breast cancer tumors, thereby enhancing tumor growth and bone metastasis [144]. Moreover, using a mathematical model, Bocci et al demonstrated that autocrine and paracrine TGFβ signaling in combination with cell-cell communication activated Notch signaling to give rise to a heterogeneous population of bCSCs, and that IL-6-enhanced Notch-Jagged1 signaling was necessary for the maintenance of bCSCs [80]. 

Other cytokines, such as tumor necrosis factor alpha (TNFα) have been shown to regulate bCSC activity. When tumor cells from the Luminal-A breast cancer subtype were exposed to TME-enriched conditions consisting of TNFα and endothelial growth factor (EGF), the breast cancer cell population became enriched for a CD44^+^CD29^+^ CSC phenotype with increased metastatic properties [83]. 

### 3.2. Immune Cells

Tumor-associated immune cells, such as natural killer cells, macrophages, neutrophils, dendritic cells, and T and B lymphocytes, relay signals to their neighboring cells through secreted cytokines. These cytokines in the TME play an important role in the development of multiple cancers. Cytokines act directly on the tumor cells, fibroblasts, and adipocytes in the TME in an autocrine or paracrine fashion regulating important cell functions. This inflamed environment in the tumor niche also effects CSC activity [84,145,146,147,148]. While CD8^+^ T cells normally play a key role in eliminating tumor cells, Santisteban et al. showed that CD8^+^ T cells can promote bCSC expansion and EMT in vivo [131]. Tumor associated macrophages (TAMs) have also been shown to play an important role in tumorigenesis [149,150]. The interaction between CD11b and Ephrin expressed by TAMs found in ERα^+^ breast cancer tumors and CD90 and Ephrin 4A (Eph4A) expressed on bCSCs results in activation of NFκB-mediated secretion of cytokines such IL-6, IL-8, and GM-CSF. These cytokines in turn play important roles in the maintenance of CSCs (i.e., self-renewal) and their proliferation and differentiation to generate new cancer cells [132].

### 3.3. Hypoxia

As solid tumors grow, due to decreased nutrient and oxygen supply, hypoxic regions develop within the TME where oxygen tension drops down to ~1%. Current evidence now indicates that this hypoxic microenvironment has the potential to regulate both normal stem cell function as well as CSC function [151,152,153,154,155,156,157,158]. It was recently demonstrated that breast cancer cells exposed to hypoxic conditions in vitro can activate the PI3K/AKT signaling pathway and promote enrichment of CD24^−^CD44^+^ CSC characteristics in xenotransplantation models [85]. in vitro studies demonstrated that repetitive cyclic exposure of normoxic and hypoxic conditions selectively enriched for breast cancer cells with a CSC phenotype. This subpopulation of cells was found to display EMT features and highly metastatic behavior in xenograft models [159]. Another study showed that a hypoxic TME resulted in hypoxia-inducible factor 1 (HIF1)-mediated expression of adenosine receptor 2B (A2BR) in human breast cancer cells. This increase in A2BR was sufficient to increase expression of CSC phenotype mediators, IL-6 and NANOG [160]. Moreover, Conley et al. reported that the use of anti-angiogenic agents such as sunitinib and bevacizumab in tumor-bearing mice created a hypoxic environment that facilitated a HIF1α mediated increase in bCSCs [161]. In addition to this, it has been shown that under chronic hypoxic conditions, the expression of HIF-2α is elevated in breast cancer cells, which in turn display a CSC phenotype by inducing the expression of stem cell markers, such as c-Myc, OCT4, and Nanog. In the same study, in vivo experiments demonstrated that increased expression of HIF-2α in breast cancer cells promotes tumorigenicity and resistance to paclitaxel via activation of Wnt and Notch signaling pathways [162]. IL-6 signaling was shown to cooperate with a hypoxic TME conditions to induce expression of C/EBPδ and other “stemness” promoting factors such as Nanog, Sox2 and Klf4 in breast cancer stem cells [163]. Lastly, a recent study demonstrated that hypoxia-induced secretion of IL6 specifically by ERα^+^ breast cancer cells was capable of elevating both ERα^+^ and ERα^−^ bCSC self-renewal and proliferation in a JAK-STAT pathway-dependent manner in vitro [164]. Based on such evidence, it is rational to hypothesize that the hypoxic areas of TME would foster the maintenance of bCSCs and that the decreased concentration of therapeutic drugs in these hypoxic areas could contribute to CSC survival and tumor recurrence.

### 3.4. Tumor Stroma

Different cell types of the stroma have been shown to play a key role in regulating CSC activity. Adipocytes secret different growth factors, cytokines, and chemokines necessary for regulation of different cellular processes such as self-renewal and proliferation [165,166,167,168]. Subcutaneous co-injection of mouse mammary adenocarcinoma cells with adipose tissue from mouse mammary fat pads resulted in increased tumor volume compared to the xenografts initiated with breast cancer cells alone [169], suggesting the importance of adipose tissue in promoting tumor growth. Furthermore, Iyengar et al., demonstrated that conditioned media from adipocytes was sufficient to enhance breast cancer cell proliferation in vitro. In addition, subcutaneous injection of breast cancer cells with murine adipocytes enhanced their tumorigenic and metastatic activity in vivo [170]. Subsequent studies demonstrated that exosomes secreted from pre-adipocytes enhanced bCSC self-renewal and breast tumorigenesis via activation of the SOX9/miR-140 signaling pathway [133], and that exosomes from mesenchymal stem cell derived adipocytes enhanced proliferation of breast cancer cells through activation of the Hippo signaling pathway both in vitro and in vivo [171]. Goto et al., also recently demonstrated that mammary gland adipocytes secrete a serine protease, Adipsin, which triggers cleavage of complement C3 and activated C3 receptor (C3aR) signaling in breast cancer cells. Inhibition of the C3a-C3aR signaling axis results in decreased proliferation and maintenance of CSC properties in human breast cancer patient derived xenograft cells [134], suggesting that the adipsin-C3a-C3aR signaling is an important component of TME and bCSC activity. 

Mesenchymal stem cells (MSCs) make up another small but important cell type present in the stroma of the mammary gland. Recently, crosstalk between MSCs and breast tumor cells has been shown [172], and accumulating evidence suggest that MSCs promote tumor growth, metastasis and development of resistance to therapy [173,174,175]. MSCs were also shown to activate P2 purinergic receptor signaling in breast cancer cells, which in turn increases their mammosphere forming ability [135]. Another study demonstrated that that ALDH1-expressing MSCs have the ability to infiltrate breast tumors and regulate bCSC self-renewal and proliferation, resulting in enhanced tumor growth in mouse xenograft models. In this model, the increase in MSC-induced self-renewal and proliferation of bCSCs was triggered by a positive feedback loop of IL-6 and CXCL7 cytokines secreted by MSCs [136]. 

In addition to adipocytes and MSCs, fibroblasts make up the majority of cells present in the stroma. Although the role of normal fibroblasts in promoting breast cancer tumor progression has been controversial, recent studies now provide evidence that both normal and activated cancer-associated fibroblasts (CAFs) can promote breast cancer cell growth in vitro and in animal model systems [176]. Indeed, these studies reveal that constitutively secreted cytokines, such as CCL7, IL-6, and IL-8, can activate the release of platelet-derived growth factor BB (PDGF-BB) from breast cancer cells that stimulates release of IL1-β by the fibroblasts and in turn induces breast cancer cell proliferation [176]. Interestingly, IL-6 and IL-8 also promote bCSC self-renewal [177,178]. Moreover, CAF-secreted prostaglandins have been shown to promote secretion of IL-6 that results in bCSC expansion [137,138]. Interestingly, senescent primary normal breast luminal cells activate breast stromal fibroblasts in an IL-8-STAT3 pathway-dependent manner. These activated fibroblasts displayed pro-carcinogenic features and promote a CSC-like phenotype by increasing expression of stem cell markers, such as CD44, ALDH, SOX2, OCT4, NANOG, and KLF4. These activated fibroblasts also induced EMT in breast cancer cells both in vitro and in vivo [179]. A recent study demonstrated that Sonic Hedgehog ligand secreted by TNBC cells confers and activates normal stromal fibroblasts. These activated fibroblasts in turn secreted FGF5 and produced fibrillar collagen-rich ECM essential for maintenance of the CSC phenotype and development of chemoresistance [180]. Another study showed that breast cancer cells activate fibroblasts and induced secretion of chemokine ligand 2 (CCL2). Fibroblast-derived CCL2 plays a key role in promoting bCSC self-renewal and tumorigenesis in a Notch1-dependent manner both in vitro and in vivo [139]. These observations indicate that the stromal fibroblasts (and in particular their activated derivatives) contribute to bCSC activity and tumorigenesis. 

Interestingly, single cell RNA-Seq analysis shows that CAFs in the TME are heterogeneous in nature and can be classified into three functionally distinct subsets based on their gene expression profiles and associated with different origins [181]. Thus, it is important to identify and characterize the subpopulation of CAFs that play a critical role in promoting bCSC activity. This will further help in using CAFs as prognostic or predictive biomarkers. Costa et al. identified four different subsets of CAFs in human breast tumors based on cell surface protein expression of fibroblast-associated protein (FAP), CD29, αSMA, fibroblast-specific protein1 (FSP1), PDGF receptor beta (PDGFRβ), and CAV1. Intriguingly, one of the CAF subsets (FAP^high^CD29^high^αSMA^high^FSP1^high^PDGFRβ^high^CAV1^low^) enhanced T-regulator cell activity in order to inhibit effector T cell proliferation, thus playing an important role in creating an immunosuppressive microenvironment in TNBCs [182]. Furthermore, Su et al. demonstrated that CAFs expressing CD10 and GPR77 were highly potent in remodeling the TME [183]. These CAFs also secrete IL6 and IL8 which can induce bCSC enrichment and chemoresistance [183]. Breast cancer cells treated with chemotherapeutic agents, such as docetaxel or cisplatin, displayed enhanced survival in the presence of CD10^+^GPR77^+^ TAFs. Furthermore, in vitro co-culturing of breast cancer cells with CD10^+^GPR77^+^ CAFs resulted in an increase in the proportion of CD24^−^CD44^+^ALDH1^+^ bCSCs and enhanced mammosphere formation. In addition, co-injection of patient derived breast cancer cells with CD10^+^GPR77^+^ CAFs promoted tumor formation as well as the proportion of bCSCs upon serial translation of breast cancer cells [183], and binding of the ECM protein hyaluronan (HA) to the stem cell receptor CD44 resulted in Nanog mediated activation of stem cell specific genes such as *Sox2* and *Rex1* in breast cancer cells. This interaction was essential in Stat3-mediated activation of multi-drug resistance (MDR1) gene expression which in turn resulted in the development of resistance to doxorubicin and paclitaxel [184]. Taken together, this evidence demonstrates the crucial role of the stromal component of the TME in bCSC maintenance and development of chemoresistance. 

## 4. Clinical Implications

Although the 10-year overall patient survival in breast cancer has dramatically improved, this disease remains the leading cause of cancer-related death in women worldwide due to tumor recurrence and therapy resistance [185]. Based on expression of receptors such as estrogen receptor (ER), progesterone receptor (PR) and HER2, breast cancers are classified clinically into luminal A (ER^+^PR^+^HER2^−^), luminal B (ER^+^PR^+^HER2^+/−^ and/or Ki67^high^), HER2 positive (ER^−^), and triple negative tumors lacking expression of all three receptors [186]. With no effective targeted therapy options currently available, triple negative breast cancer (TNBC) constitutes the most aggressive type of breast cancer, with poor overall survival. Growing evidence suggests that the aggressive nature of TNBC tumors could be due to the presence of a higher frequency of bCSCs (CD44^high^CD24^low/−^) as compared to other breast cancer subtypes [187,188,189,190]. In contrast, luminal and HER2^+^ breast cancer subtypes are thought to be ALDH^+^ (CD44^+^CD24^low/−^ALDH1^+^) [191,192]. These observations suggest that the bCSC subset within tumors is heterogeneous in nature with respect to the phenotype and possibly function among the different breast cancer subtypes. Single-cell transcriptomic analysis of primary and metastatic tumors of different breast cancer subtypes could certainly provide very interesting information about the heterogeneity of the bCSCs. Such information could then provide a framework to hypothesize as to how heterogeneity in the bCSC compartment of the different breast cancer subtypes could be predictive of therapy response and therapy resistance. 

Until recently, research efforts were mainly focused on identifying genes and genetic alterations that regulate tumor growth and progression while viewing tumors as consisting of fairly homogenous cell populations [193,194]. Such studies have led to the development of successful therapies to block signaling pathways essential to tumor growth such as estrogen receptor blockers (e.g., Tamoxifen, Fulvestrant), and the HER2 receptor blocker, Herceptin. However, the current clinical challenge in the management of breast cancer is the development of therapy resistance, relapse, and metastasis. To this end, bCSCs have now been established to be the cells responsible for maintaining and regenerating tumors. However, targeting bCSCs has proven to be challenging, as most of the current therapy options fail to target these cells, which remain protected in their niche and contribute to therapy resistance, relapse, and ultimately metastasis. To this end, regulation of bCSC function and induction of chemoresistance by external factors such as cytokines, chemokine and hypoxia is becoming apparent as potential strategies that could target the interaction of bCSCs with cellular and non-cellular components of the TME as more effective therapeutic approaches. For example, the CXCR1 inhibitor repertaxin in combination with lapatinib significantly abrogated bCSC activity in both HER2-positive and negative tumors in preclinical animal models [129,195]. Treatment with an IL6-neutralizing antibody completely eliminated chemoresistance of ovarian and lung cancer stem cells in vivo [196,197], suggesting that targeting IL-6 could be an effective strategy in eliminating bCSCs as well. In glioma, the TGFβ inhibitor SB431542 promoted differentiation of CSCs in vitro [198]. These observations suggest that targeting the secreted factors of the TME could represent an effective therapeutic strategy in eliminating bCSCs. It is, however, important to take into consideration the subtle differences between mouse and human mammary gland. For preclinical trials, it would important to use an orthotopic mouse model whose tissue environment has been altered (humanized) to resemble the human breast. Moreover, reconstruction of the tumor microenvironment in 3D Matrigel in patient-derived organoid cultures (PDOCs) would be another way of assessing the potential action of these neutralizing antibodies or inhibitors on bCSCs. The strength of the PDOC system is that it enables the study of tumor-infiltrating leukocytes in bCSC function which is not easy to model in the mouse.

## 5. Conclusions

The postnatal adult mammary gland is maintained by the most primitive self-renewing population of MaSCs. Depending on microenvironmental cues, MaSCs differentiate to lineage-restricted progenitors, which eventually generate mature luminal and myoepithelial cells of the mammary gland. This defines the hierarchical organization of mammary epithelial cells and ultimately the normal mammary gland. Studies performed on mouse models have demonstrated that both intrinsic factors (effectors of Notch and Wnt signaling pathways) and extrinsic factors (tissue microenvironment) regulate MaSC function. Multiple studies have identified surface markers to isolate and characterize the murine MaSC population; however, there are still a lack of defined unique stem cell markers to definitively identify MaSCs in the human breast. As a result of this, the activity of MaSCs and their interactions with the microenvironment in human breast tissue remains poorly understood. Importantly, bCSCs share many of the specific characteristics and functions of normal MaSCs (summarized in Figure 2), including their dependence on the tissue or tumor microenvironment for regulating their proliferative and self-renewal potentials. Accumulating evidence show that molecular mechanisms that regulate normal MaSCs (such as Notch and Wnt signaling pathways) can also help in maintenance of bCSCs’ phenotype, resulting in breast tumorigenesis, progression, and metastasis. This suggests that understanding the role of normal breast MaSCs and their tissue environment would provide some insights into understanding the role of TME in regulating breast CSC activity.

Despite early detection and therapy options, the majority of deaths in breast cancer patients occur due to resistance to therapy and metastasis. bCSCs represent a small number of cells within the heterogeneous tumor cell populations that have the potential to regenerate the tumor and are thought to be responsible for tumor recurrence, therapy resistance and ultimately metastasis. Our previous notion that breast cancer manifestation occurs solely by cell intrinsic factors (mutations, gene amplification) has been challenged by more recent studies described in this review and elsewhere. Extensive research combined with clinical trials has been conducted with the goal of eliminating CSCs, however due to CSC plasticity, these attempts have not been very successful. There is compelling evidence that the tumor microenvironment plays a critical role in regulating CSC plasticity that drives the ability of a non-CSC to dedifferentiate into a CSC, thereby contributing to tumor initiation, progression, therapy resistance, and metastasis. Studies have shown that a bCSC-supportive niche consisting of activated fibroblasts, immune cells and adipocytes alter bCSC activity either by direct interaction or through secreted factors. Several preclinical trials targeting cytokines have shown promising results in inhibiting tumor growth. Taken together, these studies suggest that targeting the cellular and non-cellular components of the tumor microenvironment could serve as an effective therapeutic strategy for both reducing tumor growth and also sensitization of the therapy resistant bCSCs.

## Figures and Tables

**Figure 1 cancers-11-01240-f001:**
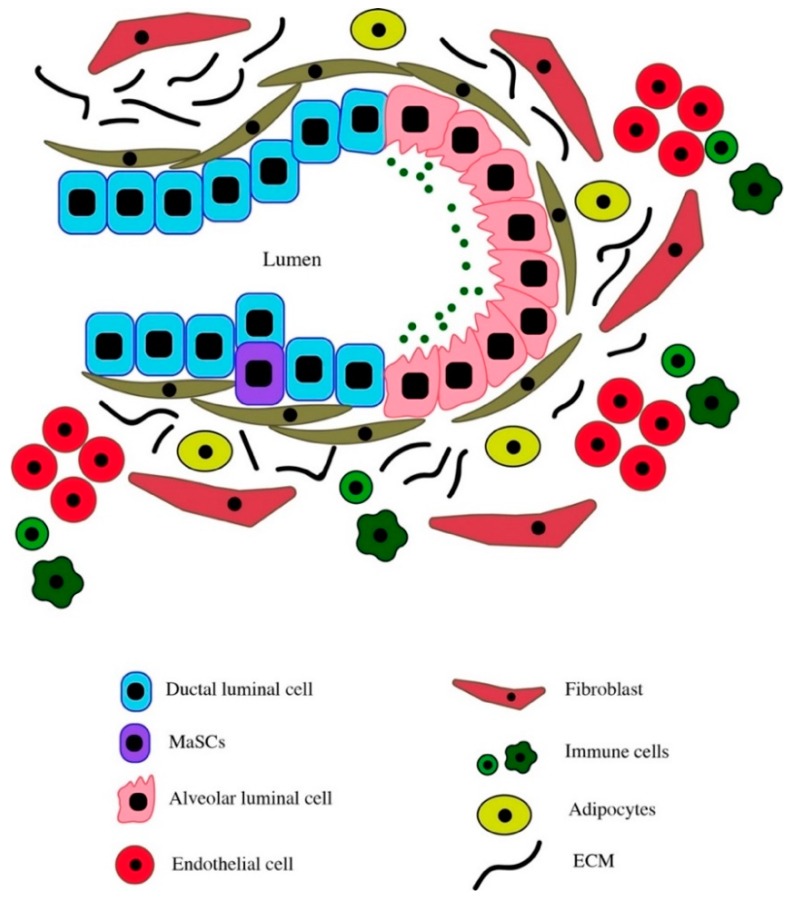
Schematic representation of the breast tissue microenvironment showing bilayered arrangement of ducts and alveolar structures consisting of luminal and myoepithelial cells. These structures are embedded in fibrous stroma consisting of fibroblasts, adipocytes, and immune cells. The luminal cells in the ducts are surrounded by a continuous lining of myoepithelial cells, while the myoepithelial cells lining the luminal cells in alveoli are discontinuous, which allows luminal cells to interact with surrounding stroma.

**Figure 2 cancers-11-01240-f002:**
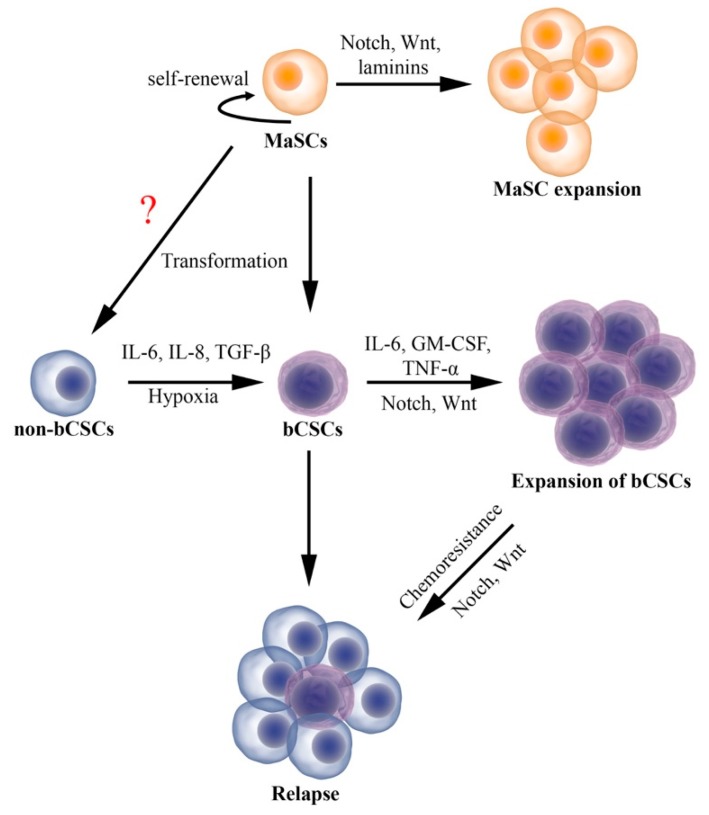
Schematic representation of mammary stem cell (MaSC) expansion, breast cancer stem cell (bCSC) plasticity, development of chemoresistance and relapse. Different cytokines released by immune cells, fibroblasts, adipocytes along with tumor cells in the tumor microenvironment regulate both MaSC and bCSC activity. It is possible that MaSCs could acquire sufficient genetic changes that allow them to transform directly into malignant cancer cells devoid of stem cell properties (non-bCSCs). However, this hypothesis requires further experimental evidence.

**Table 1 cancers-11-01240-t001:** Summary of the role of cytokines, immune cells, and stromal cells in regulating breast cancer stem cell (bCSC) activity in the tumor microenvironment.

Stimulant	Action	References
Interleukin-6	• Dedifferentiation of CD44^low^ MCF10A to CD44^high^ cells	[125]
	• Activation of JAK1/STAT3 signaling pathway in TNBC cell lines	[126]
	• Activation of JAG1-NOTCH3 signaling pathway in ER^+^ breast cancer cell lines	[127]
Interleukin-8	• Enhances bCSC activity and induction of chemoresistance in TNBC cells	[128]
	• Regulation of bCSCs in HER2+ breast cancers via activation of IL-8-CXCR1 signaling axis	[129]
TGFβ	• Increases the number of CD44^high^ CD24^low^ cell population	[130]
TNFα	• Enriches the CD44^+^CD29^+^ bCSC population in Luminal-A breast cancer cells	[83]
Oncostatin-M	• Upregulation of SNAIL and CD44 expression in TNBC cell lines	[86]
	• Enhances tumor forming ability of TNBC cells	[86]
CD8^+^ T cells	• Promotes bCSCs expansion and EMT	[131]
TAMs	• Promotes secretion of cytokine such as IL-6, IL-8 and GM-CSF and maintenance of bCSCs	[132]
Stromal Cells ⚬ **Pre-adipocytes** ⚬ **Adipocytes** ⚬ **MSCs** ⚬ **CAFs**	• Enhances bCSC self-renewal via exosome secretion	[133]
	• Secretes adipsin and enhances bCSC activity through activation of C3a-C3aR signaling	
	• Increases mammosphere-forming ability of breast cancer cell via activation of P2 purinergic pathway	[134]
	• Secretes IL-6 and CXCL7 and enhances bCSC self-renewal and proliferation in mouse xenograft model	[135]
	• Secretes prostaglandin and enhances bCSC expansion	[136]
	• Promotes bCSC self-renewal via CCL2 secretion	[137,138]
	• Secretes IL-6 and IL-8 thereby protects bCSCs from chemotherapeutic agents	[139]

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
