# Peer review of "Role of the Microenvironment in Regulating Normal and Cancer Stem Cell Activity: Implications for Breast Cancer Progression and Therapy Response"

_cancers, 2019, doi:10.3390/cancers11091240_

Round 1

Reviewer 1 Report

This review is well-written. However, if the authors rewrote it in accordance with the following comments, the reviewer believe that this manuscript get better and will be suitable for publication in our journal

Section 2 is verbose and is not easy for the readers to read. The reviewer suggest that the authors use subheading like Section 3. In Figure2: The author described that MaSC directly transformed into breast non-CSC cancer cells. The reviewer suggest that the author show the references where this phenomenon is demonstrated. MaSCs as well as mammary progenitor cells can transform into breast CSC. The authors had better describe it and quote the references.

Author Response

Reviewer 1:

"This review is well-written. However, if the authors rewrote it in accordance with the following comments, the reviewer believe that this manuscript get better and will be suitable for publication in our journal."

"i- Section 2 is verbose and is not easy for the readers to read. The reviewer suggest that the authors use subheading like Section 3".

Response: We thank the reviewer for this suggestion. Section 2 has now been reorganized and broken down with subheadings to improve readability.

ii- In Figure2: The author described that MaSC directly transformed into breast non-CSC cancer cells. The reviewer suggest that the author show the references where this phenomenon is demonstrated. MaSCs as well as mammary progenitor cells can transform into breast CSC. The authors had better describe it and quote the references.

Response: We thank the reviewer for bringing this issue to our attention, it was an oversight on our part. The ability of MaSCs as well as mammary progenitor cells to transform into breast CSC (bCSCs) has now been discussed in the article and appropriate references have been cited (Lines 203-210 in the revised manuscript). With regards to the differentiation of MaSCs into non-bCSC cancer cells, this issue although plausible, has not been demonstrated experimentally. To clarify this, we have now placed a question marks on the arrow connecting MaSCs to the non-bCSCs cancer cells and added a sentence to the legend to explain this.

Reviewer 2 Report

Bhat et al. explains the crosstalk between mammary epithelial cells and their tissue microenvironment. This review focuses on the role of the microenvironment into the homeostasis of the mammary gland epithelium and tumorigenesis.

In general, the review is well constructed and acknowledgeable. Each section has a distinct purpose and overall the topic is very interesting. The authors have covered all the components of the microenvironment that interact with breast epithelial cells, from cytokines to specific cell types.

Although this manuscript is complete and the figures are well designed, I have a minor comment:

In Section 2, and later in Section 5, the authors mention MaSCs, which were described more than ten years ago, based on transplantation assays and have ignored all the new evidences based on lineage tracing studies that show the lack of multipotent stem cells in murine adult glands. It has been extensively demonstrated by different groups that the maintenance of the mammary epithelium in adult animals is driven by basal and luminal unipotent stem cells that, although having different capabilities to generate outgrowths in cleared fat pads, both are responsible for maintaining their own compartments. Indeed, there are indications that show how during the embryonic development such multipotency is lost. With this, I am saying that the concept of MaSC is obsolete (regarding murine mammary glands). Reffs: Van Keymeulen et al. 2011; Wuidart et al. 2016; Wuidart et al. 2018; Lilja et al. 2018; and Chung CY et al. 2019, just to mention some of a long list.

Since the purpose of this review is not discussing the cellular hierarchy of the mammary gland, I will not insist in removing this part, but I encourage the authors to read those articles. However, in line with their message, it would be interesting to mention that different mammary compartments (luminal vs basal) have different capacities to repopulate a cleared fat pad, and that is probably due to the fact that basal cells are used to interact with the stroma, contrary to the luminal cells, which grow worse in transplantation assays. This point could be discussed/or left as an open question somewhere.

Author Response

Reviewer 2:

"Although this manuscript is complete and the figures are well designed, I have a minor comment: In Section 2, and later in Section 5, the authors mention MaSCs, which were described more than ten years ago, based on transplantation assays and have ignored all the new evidences based on lineage tracing studies that show the lack of multipotent stem cells in murine adult glands. It has been extensively demonstrated by different groups that the maintenance of the mammary epithelium in adult animals is driven by basal and luminal unipotent stem cells that, although having different capabilities to generate outgrowths in cleared fat pads, both are responsible for maintaining their own compartments. Indeed, there are indications that show how during the embryonic development such multipotency is lost. With this, I am saying that the concept of MaSC is obsolete (regarding murine mammary glands). Reffs: Van Keymeulen et al. 2011; Wuidart et al. 2016; Wuidart et al. 2018; Lilja et al. 2018; and Chung CY et al. 2019, just to mention some of a long list.

Since the purpose of this review is not discussing the cellular hierarchy of the mammary gland, I will not insist in removing this part, but I encourage the authors to read those articles. However, in line with their message, it would be interesting to mention that different mammary compartments (luminal vs basal) have different capacities to repopulate a cleared fat pad, and that is probably due to the fact that basal cells are used to interact with the stroma, contrary to the luminal cells, which grow worse in transplantation assays. This point could be discussed/or left as an open question somewhere".

Response: As pointed out by the reviewer, discussions of the mammary stem cell hierarchy falls outside of the scope of this article. However, it is very interesting to note that the normal mammary gland development field remains divided based on the presence of bipotent and unipotent stem cells. Since this review article is focused on the role of microenvironment in regulating normal and cancer stem cells in the breast tissue, we chose to take an unbiased stance and hence did not discuss the heterogeneity of the mammary stem cells in detail. However, as per reviewer’s suggestion, we have now added a brief discussion regarding the current evidence in support of unipotent stem cells in the mouse mammary gland (Lines 94-97 in the revised manuscript). However, the presence of such unipotent stem cells in human breast tissue has not been shown.